# Gold Nanoparticles Modification with Liquid Crystalline Polybenzylic Dendrons via 1,3-Dipolar Cycloaddition

**DOI:** 10.3390/nano12224026

**Published:** 2022-11-16

**Authors:** José Antonio Ulloa, Joaquín Barberá, José Luis Serrano

**Affiliations:** 1Departamento de Polímeros, Facultad de Ciencias Químicas, Universidad de Concepción, Casilla 160-C, Calle Edmundo Larenas 129, Concepción 4070371, Chile; 2Departamento de Química Orgánica, Facultad de Ciencias, Instituto de Nanociencia y Materiales de Aragón (INMA), CSIC-Universidad de Zaragoza, 50009 Zaragoza, Spain

**Keywords:** gold nanoparticles modification, liquid crystalline dendrons, click-chemistry, nanoparticles functionalization

## Abstract

A series of six polybenzylic dendrons with an alkynyl focal point were synthesized for their incorporation to gold nanoparticles. Five of these compounds showed columnar mesomorphism in a wide range of temperatures. These dendrons were reacted with gold nanoparticles stabilized with a combination of a dodecanethiol and 11-azidoundecane-1-thiol. The azido group of the last compound allowed the functionalization of the nanoparticles with the six polybenzylic dendrons by 1,3-dipolar cycloaddition between their alkynyl groups and the terminal azido groups of the thiols. A high efficiency of the cycloaddition process (47–69%) was confirmed by several experimental techniques and no decomposition or aggregation phenomena were detected in the dendron-coated nanoparticles. The involved mechanism and the resulting percentage composition of the final materials are discussed. The results of the ulterior growth of the nanoparticles by thermal treatment are influenced by the size and the shape of the dendron and the temperature of the process. The structures of the final nanoparticles were investigated by TEM, DSC, TGA, NMR and UV-Vis spectroscopy. These nanoparticles do not show liquid crystal properties. However, a melting process between a crystalline and a fluid phase is observed. In the solid phase, the nanomaterials prepared show a short-range interaction between nanoparticles with a 2D local hexagonal order. A near-field effect was observed in the UV-vis spectra by coupling of different surface plasmon resonance bands (SPR) probably due to the short-range interactions. The main novelty of this work lies in the scarcity of previous studies of gold nanoparticles coated with dendrons forming themselves columnar mesophases. Most of the studies reported in the literature deal with gold nanoparticles coated with calamitic mesogens. Additionally, the effect of the thermal treatment, which in a previous paper was shown to increase the mean size of the nanoparticles without increasing their size polydispersity, has been studied in these materials.

## 1. Introduction

Liquid crystal dendrimers, also known as *dendromesogens*, were described for the first time by Percec et al. [1] and shortly later by Shibaev and coworkers [2]. In some cases, the dendrimer includes mesogenic units in its molecular structure, whereas in other cases the liquid crystal behavior is obtained without the presence of mesogenic units. In this second case, the mesomorphic behavior can be produced by the microsegregation phenomenon observed between incompatible regions of the molecules through an appropriate balance between enthalpy and entropy in such a way that the molecules can adopt an anisotropic shape. Percec reported this phenomenon in 2004 for a type of polybenzylic dendritic molecule that contains aliphatic chains in its periphery and that does not present any mesogenic moiety [3,4,5]. In addition to phase microsegregation by selective interactions between the aromatic regions on one side and between the aliphatic regions on the other side, these molecules can interact with each other to generate supramolecular species with shapes, such as rods, wedges, half-disks, disks, cones, etc., able to produce specific mesomorphism (calamitic, columnar, cubic, etc.) [6,7,8,9].

Besides this, some dendrimers have been proved to be useful to functionalize metal nanoparticles. For this purpose, dendrimers can be used as templates in the nanoparticle synthesis or can be used to coat the nanoparticles as stabilizing agents [10,11,12,13,14,15,16]. In the last case, the dendrimers functionalize the nanoparticle surface and, in this way, provide appropriate control of size and shape, avoiding the formation of macroscopic aggregates and producing new physical properties if the dendrimer contains active groups. In addition to this, in a recent article we have given evidence that these bulky ligands avoid the paramagnetic behavior observed in aggregated nanoparticles [17]. For the synthesis of dendrimer-coated nanoparticles, three approaches can be employed: (i) In the direct method, the reduction of a metal salt is carried out in the presence of the stabilizing ligand, i.e., the dendrimer or dendron of interest. The advantage of this method is the high functionalization degree, but the drawback is the limited control of size and shape of the resulting nanoparticle [18,19,20,21,22,23,24,25]. (ii) The nanoparticle can initially be coated with a stabilizing ligand and then this ligand is displaced by the ligand of interest to yield nanoparticles coated with both types of ligand. The main drawback of this method is that considerable excess of the second ligand is needed and sometimes an appropriate control of size and shape is difficult [26,27,28,29]. (iii) The third strategy is based on the synthesis and stabilization of the nanoparticle with bifunctional ligands, which are linked to the nanoparticle surface by one of their functional groups, whereas the other one remains available to react with other molecules, for instance, dendrons if the goal is to coat the nanoparticle with dendritic molecules. The main advantage of this procedure is that the nanoparticle size is not significantly affected in subsequent reactions and that the nanoparticle surface can be highly functionalized without needing an excess of reactants [30,31,32,33].

In the field of nanomaterials, it is desirable to achieve a good control of size, shape and chemical composition to ensure appropriate control of the physical and chemical properties [34,35]. The use of liquid crystals is a very favorable approach to achieve these goals [31,32,33,36,37,38,39,40,41,42,43]. For this purpose, it is necessary to synthesize nanoparticles as monodisperse as possible. To this end, thermolitic methods consist of the application of a thermal stimulus on the coated nanoparticles with the aim of producing nucleation processes that lead to the controlled growth of the metallic core [44,45,46]. In general, the procedure involves the thermal treatment of the nanoparticles at a given temperature for a period of time. The chosen temperature plays a crucial role in the extent of growth and the homogeneity of the final nanomaterials.

In this report the synthesis, chemical stabilization, and characterization of gold nanoparticles with controlled size by means of two steps are described: (i) In the first one, dodecanethiol-coated gold nanoparticles (AuDT) were functionalized via ligand exchange, with 11-azidoundecane-1-thiol (AuDT-AT). Then, terminal azido groups were reacted via 1,3-dipolar cycloaddition, with mesogenic dendrons containing an alkynyl group at their focal point. (ii) In a second step, the nanoparticles obtained by this method were submitted to an isothermal treatment at controlled temperatures and times, and the resulting properties were investigated. This method has been tested in a previous work, showing its ability to generate nanoparticles of bigger size and low polydispersity [29].

Additionally, the effect of the mesomorphic character of the dendritic ligands on the tendency of the nanoparticles to adopt regular arrangements was studied. Attaching liquid crystalline molecules to nanoparticles is promising for the design of functional materials than can be manipulated by external stimuli, such as temperature, light and electric and magnetic fields. This work intends to be a proof of concept with the aim to explore the possibilities of modifying the interactions between the nanoparticles and modulating their properties by coating their surface with mesogenic dendrons.

## 2. Materials and Methods

### 2.1. Reagents and Equipment

The materials and methods are included in Appendix A. For abbreviations see Appendix A.

All reagents and silica gel were purchased from Sigma-Aldrich^®^. Anhydrous THF (St. Louis, MO, USA) and DCM were purchased from Scharlab (Barcelona, Spain) and dried using a solvent purification system.

Size exclusion chromatography (SEC) was performed with Sephadex LH-20 purchased from GE Healthcare Life Science^®^ (Marlborough, MA, USA).

FTIR spectra were obtained using a Thermo NICOLET Avatar 360 FT-IR spectrophotometer and Bruker Vertex 70 MKII Golden Gate Single Reflection ATR System.

NMR experiments were carried out on a Bruker AV-400 spectrometer operating at 400 MHz for ^1^H and 100 MHz for ^13^C equipped with a QNP probe (Quattro nucleus probe).

MALDI-TOF MS was performed on an Autoflex Mass Spectometer Bruker Daltonics apparatus using ditranol as matrix.

X-ray photoelectron spectra (XPS) were recorded on a Kratos Axis Supra XPS system with a base pressure of 1 × 10^−10^ mbar using a monochromated Al Kα X-ray source. XPS survey scans were taken at a pass energy of 225 W (15 mA/15 kV). Data analysis was done with the CASA XPS software package.

Thermogravimetric analysis (TGA) was performed using a Q5000IR apparatus from TA Instruments.

Thermal transitions were determined by differential scanning calorimetry (DSC) using a Q2000 equipment from TA Instruments.

Mesogenic behavior was investigated by polarized-light optical microscopy (POM) using an Olympus BH-2 polarizing microscope fitted with a Linkam THMS600 hot stage.

X-ray diffraction (XRD) was performed with an evacuated pinhole camera (Anton-Paar) operating a point-focused Ni-filtered Cu-Kα beam. Powdered samples of the alkyne focal-point dendrons were placed in Lindemann glass capillaries (0.9 mm diameter). The patterns were collected on flat photographic film perpendicular to the X-ray beam. An Anton-Paar high-temperature attachment was used when necessary.

Transmission electronic microscopy (TEM) images and selected area electron diffraction (SAED) patterns were obtained by FEI Tecnai TF20 in a 200 kV FEG high resolution Transmission Electron. Data analysis was done with the Image-J software package.

UV-Vis absorption spectroscopy was performed using an ATI-UNICAM UV4-200 spectrophotometer with quartz cuvette 10 mm pathlength.

### 2.2. Synthesis of Alkyne Focal-Point Fréchet-Type Dendrons

The synthesis of the precursory Fréchet-like polybenzyl alcohol dendrons was previously described [47,48]. The alkyne function was incorporated following the Steglich esterification procedure between the polybenzylic alcohol and propiolic acid, yielding alkyne-focal point dendrons 1–6 (Figure 1).

The details of the synthesis and structural characterization of all the functionalized alkyne focal-point dendrons and their precursors are collected in Appendix A. All the intermediate dendrons were characterized by nuclear magnetic resonance (^1^H-NMR, ^13^C-NMR), Fourier-transform infrared spectroscopy (FTIR) and mass spectrometry (EM-MALDI). For a representative example, see Appendix A.

### 2.3. Synthesis of AuDT, AuDT-AT and AuDT-AT-TA@L3-3,4,5 Gold Nanoparticles

#### 2.3.1. Synthesis of AuDT Pristine Nanoparticles

The gold nanoparticles stabilized with dodecane-1-thiol (AuDT) were synthesized by the method of Brust et al. (See Appendix A) [29,49]. Scanning transmission electron microscopy (STEM) experiments show spherical gold nanoparticles with a low size dispersity and mean diameter values of 2.0 ± 0.4 nm, calculated from a statistically representative number of particles. For each TEM specimen, at least 200 to 250 particles were measured using several microphotographs obtained from different areas of the grid.

#### 2.3.2. Synthesis AuDT-AT Pristine Nanoparticles by Ligand Exchange Reaction of the AuDT Nanoparticles with the 11-Azidoundecane-1-thiol (AT)

The ligand 11-azidoundecane-1-thiol (AT) was prepared by a modification of the procedure described by Collman et al. [50] (see Section 2.2, Appendix A).

To introduce 11-azidoundecane-1-thiol (AT) in these nanoparticles, the procedure reported by Astruc and coworkers was followed [51]. A solution in dichloromethane of AuDT and AT in the appropriate proportion was stirred at room temperature in an argon atmosphere. Most of the solvent was evaporated and the concentrated solution was poured on ethanol, stirred and cooled at −18 °C. After several centrifugations to remove the non-reacted ligand; the dry product was purified by size exclusion chromatography (SEC). The experimental details are described in Section 2.3.1 and Appendix A.

#### 2.3.3. Synthesis of the AuDT-AT-TA@L3-3,4,5 Gold Nanoparticles by Functionalization of AuDT-AT Nanoparticles with Alkynyl Dendritic Ligands by Huisgen 1,3-Dipolar Cycloaddition

The reaction was carried out by mixing a solution of the appropriate proportion of AuDT-AT nanoparticles and the corresponding alkynyl ligand in THF with a solution of CuSO_4_ and sodium L-ascorbate in water, tris(benzyltriazolyl)methyl amine (TBTA) and DMF, and the resulting solution was stirred at 30 °C. After pouring on a 1:1 mixture of dichloromethane and 15 N ammonium hydroxide, the organic phase was dried and evaporated, and the product was purified with Sephadex LH-20^®^. The experimental details are described in Section 2.3.2 and Appendix A.

## 3. Results and Discussion

### 3.1. Characterization of the Optical, Thermal, Thermodynamic Properties and Mesogenic Behavior of Alkyne Focal-Point Fréchet-Type Dendrons

Thermal stability of the dendrons synthetized was studied by thermogravimetric analysis (TGA). All the alkyne focal point compounds 1–6 showed good thermal stability. Thus, the 5% mass loss temperature (T5%) in all samples occurred at temperatures above 270 °C, a value significantly higher than the isotropization temperature (temperature of the transition to the isotropic liquid, see Table 1 and Section 3.1 and Appendix A).

By polarized optical microscopy (POM) observations, using a heating stage, it was possible to detect the existence of liquid crystal phases. Compounds 1(L2-3,4), 2(L4-3,4) and 5(L6-3,4,5) show enantiotropic mesomorphism (liquid crystal behavior was observed both on heating and cooling), whereas compounds 4(L3-3,4,5) and 6(L9-3,4,5) show monotropic behavior mesomorphism (liquid crystal behavior was observed only on the cooling process). On the other hand, compound 3(L4-3,5) does not show liquid crystal behavior (see Table 1). Focal-conic textures that agree with columnar mesophases were easily identified in compounds 1(L2-3,4), 2(L4-3,4), 4(L3-3,4,5) and 5(L6-3,4,5). In compound 6(L9-3,4,5) birefringent textures, not easily identified by POM, were observed within a temperature range between the highly-fluid isotropic liquid and the solid crystalline phase. This fact supports the existence of mesomorphism. Some representative textures are shown in Appendix A).

Thermal and thermodynamic parameters of the phase transitions observed were measured by differential scanning calorimetry (DSC). All the DSC curves are reproducible after the first heating-cooling cycle and no-decomposition process was observed (DSC data are collected in Table 1).

The information drawn from the POM observations and the DSC measurements was completed with the information obtained by X-Ray diffraction in the liquid crystalline phase (Table 1). Except for the mesophase of 1(L2-3,4), which was studied at 55 °C; the X-ray diffractograms were recorded at room temperature after a thermal treatment consisting of heating up to the isotropic liquid and cooling down to room temperature. Additionally, the X-ray data allow an estimation of the number of molecules per unit cell (see Section 3.3, Appendix A).

The X-ray studies confirm the existence of a columnar hexagonal phase in compounds 2(L4-3,4), 4(L3-3,4,5) and 5(L6-3,4,5). In a columnar hexagonal mesophase the molecules stack in columns and the columns adopt a hexagonal packing. Fortunately, the monotropic mesophase of 4 is kinetically stable at room temperature for a time long enough for its characterization by X-ray diffraction. The compound 1(L2-3,4) shows a columnar rectangular phase. In a columnar rectangular mesophase the columns adopt a rectangular (orthorhombic) packing. On the other hand, compound 6(L9-3,4,5) crystallizes in the conditions of the X-ray experiments. These compounds are an example of dendritic compounds with liquid crystal properties that do not include mesogenic units in their molecules. The existence of mesomorphism is due to the presence of chemically different segments that favor microsegregation in amphipathic blocks that produce an anisotropic arrangement. This arrangement is based on the columnar stacking of disk-shaped aggregates that contain a variable number of molecules. Depending on the column diameter, on the size of the dendron and on the number of its hydrocarbon chains, the number *Z* of molecules per aggregate (molecules necessary to fill the column cross-section, see Table 1) evolves from *Z* = 12 for 1(L2-3,4) (the dendron with the lowest number of chains) to *Z* = 4 for 6(L9-3,4,5) (the dendron with the highest number of chains).

From the data gathered in Table 1, it can be concluded that the number and position of the hydrocarbon chains plays a fundamental role in the mesomorphic behavior. A low number of chains (two chains) produces Col_r_ mesomorphism (compound 1(L2-3,4)), whereas four or more chains generate Col_h_ mesomorphism (rest of the compounds). Moreover disubstitution in positions 3,5 precludes mesomorphism (compound 3(L4-3,5)), whereas substitution in positions 3,4 or 3,4,5 favors mesomorphism (rest of the compounds).

Absorption spectra of the alkyne-focal dendrons were recorded from 10^−5^ M solutions in chloroform. For the emission spectra, 10^−6^ M solutions were excited at 241 nm. The results of these studies are collected in Appendix A). All compounds show an absorption band at approximately 275–283 nm and a structured emission band at approximately 398–410 nm. Curiously, the compounds with the lowest and the highest numbers of chains present the most intense emission bands. This section may be divided by subheadings. It should provide a concise and precise description of the experimental results, their interpretation, as well as the experimental conclusions that can be drawn.

### 3.2. Characterization of AuDT and AuDT-AT Gold Nanoparticles

The IR spectra of the nanoparticles confirmed the incorporation of the ligand by the appearance of an intense band at 2088 cm^−1^, corresponding to the azido group (Section 2.3.1 and Appendix A). Likewise, the ^1^H-NMR spectra (see Figure 1) show the appearance of the signal labelled **a** at 3.25 ppm from the methylene group in alpha position to the azido group (t, -CH_2_-N_3_). The signals **b_1_** and **b_2_** corresponding to the methylene groups in alpha to the thiol group in each ligand appear superimposed at 2.68 ppm. From the relative areas of **a**, **b_1_** and **b_2_** peaks (area **a** = area **b_2_**;≥ area **b_1_** = area **b_1-2_**-area **a**) it was concluded that the ligand composition in the nanoparticles was 17% DT and 83% AT (see Table 2). Scanning transmission electron microscopy (STEM) experiments also show spherical gold nanoparticles with a low size dispersity and mean diameter values of 2.2 ± 0.4 nm (See Figure 1b,c and Table 2), very similar to the values observed in the starting material (AuDT, 2.0 ± 0.4 nm).

UV-Vis spectroscopy studies show that after ligand exchange there are no surface plasmon resonance bands, which rules out macroscopic aggregation in the material.

### 3.3. Characterization of AuDT-AT-TA@Ln Gold Nanoparticles

The nanoparticles incorporating the dendrons are denoted AuDT-TA@Ln, where TA is the triazole formed by click reaction and Ln the dendritic ligand incorporated by copper-catalyzed cycloadditions. The AT acronym has been maintained in the nomenclature because the AT chains react only partially, forming the triazole rings and, as a consequence, some AT ligands remain in the nanoparticle (see Figure 2).

The ^1^H NMR spectra of the all gold nanoparticles prepared are gathered in Appendix A). As an example, the spectra of the L3-3,4,5 alkyne-focal point ligand and the AuDT-AT and AuDT-AT-TA@L3-3,4,5 gold nanoparticles are represented in Figure 3 (IR) and Figure 4 (^1^H NMR). As can be seen in Figure 3b the AuDT-AT nanoparticles exhibit the band characteristic of the azido group at 2090 cm^−1^, whereas the alkynyl dendrons exhibit the stretching vibrations of the carbon–carbon triple bond and of the Csp-H bond at 2120 cm^−1^ and 3265 cm^−1^, respectively (Figure 3a). After the Huisgen 1,3-dipolar cycloaddition (Figure 3c), the band of the azido group, reduced drastically its intensity and logically the signals at 3265 and 2120 do not appear. Moreover, the bands characteristic of the groups of the dendron, such as aromatic rings, benzyl ethers and ester groups, emerged in the region between 1700 and 1100 cm^−1^.

The average size of the nanoparticles was measured by transmission electron microscopy (TEM) and the results are gathered in Table 2. It is observed that the size and shape of the nanoparticles do not change significantly upon cycloaddition, which means that aggregation processes do not take place. As an example, in Figure 3d,e are represented the TEM micrograph and the NP’s size distribution histogram for the AuDT-AT-TA@L3-3,4,5 nanoparticles. As mentioned above, the histogram was obtained from a statistically representative number of particles, using several microphotographs obtained from different areas of the grid.

The ^1^H-NMR spectroscopy of the AuDT-AT-TA@L3-3,4,5 (See Figure 4c) reveals the appearance of the signal from the proton **d** of 1,2,3-triazole at 8.05 ppm (s, 1H) and a triplet signal **c** at 4.39 ppm (t, 3H) corresponding to the methylene group neighbor to the triazole heterocycle. However, a noticeable decrease in the intensity of the signal from the methylene group **a** adjacent to the azido group is observed. Some shifting of several signals is also observed as a result of the changes in the chemical environment, for instance the signals from the aromatics protons.

The signals of the methylene protons next to the sulfur atom for the three ligands: **b_1_** for DT, **b_2_** for AT and **b_3_** for TA@L3-3,4,5 appear superimposed (signal **b_1-3_**). However, the area of the signal **a** = area of the signal **b_2_** reports the relative proportion of the AT ligand and the area of the signal **c** = area of the signal **b_3_** reports the relative proportion of L3-3,4,5. Consequently, the relative proportion of ligand DT is obtained from the equation area of **b_1-3_** = area of **b_1_** + area of **a** + area of **c**. The resulting proportions of ligands in the nanoparticles studied are gathered in Table 2.

The study of the ligand ratio in the different nanoparticles allows us to obtain some conclusions about the Huisgen 1,3-dipolar cycloaddition reaction. Thus, it is possible to observe that the reaction with the alkynyl dendrons L2-3,4, L4-3,4, L3-3,4,5 and L6-3,4,5 does not significantly modify the proportion of DT ligands in the nanoparticle, which indicates that there is no loss of ligands in this process. On the other hand, in compounds L4-3,5 and L9-3,4,5 an increase in the DT ligand percentage is observed. A plausible explanation for this is the loss of some AT ligands in the cycloaddition reaction. Particularly remarkable is this effect in the larger ligand L9-3,4,5. Assuming that the DT ligands remain in the nanoparticle (approximately 17% data, in brackets, in Table 2), a drastic reduction is deduced for the amount of AT and TA@L9-3,4,5 ligands to 13% and 30%, respectively (in brackets), indicating a loss of a significant number of these ligands (approximately 50% of AT ligands are lost). The same calculation performed for the cycloaddition reaction with dendron L4-3,5 suggests a loss of approximately 35% of AT ligands (Table 2). The percentage of functionalized AT groups after the cycloaddition reaction varies between 47–48% observed for dendrons L4-3,5 and L6-3,4,5 and 69% observed for dendron L9-3,4,5.

The study of the nanoparticles by POM and DSC revealed the absence of mesomorphism in these materials. Thus, the liquid crystal properties of the bare dendrons are not maintained in the nanoparticles. Nevertheless, a phase transition between a crystal phase and a fluid phase has been observed in all nanomaterials prepared. Curiously the temperature of this transition decreases as the number of the terminal chains increases: TA@L3-3,4,5 (90 °C) > TA@L6-3,4,5 (57 °C) > TA@L9-3,4,5 (46 °C) and TA@L2-3,4 (86 °C) > TA@L4-3,4 (67 °C). The nanoparticles bearing the L4-3,5 dendron exhibit the lowest temperature value for this transition: TA@L4-3,5 (28 °C).

UV-vis absorption spectroscopy studies show that, after cycloaddition, there is no evidence of surface plasmon resonance bands, which rules out macroscopic aggregation in the material (Appendix A). With regard to the fluorescence emission (Appendix A), also in this case, as occurs in the dendronic precursors, a structured band is observed. These bands show a batochromic shift of approximately 50 nm compared to the alkynyl dendrons. This fact can be due to the generation of the 1,2,3-triazole aromatic ring adjacent to the carbonyl group, which extends electron conjugation.

X-ray photoelectron spectroscopy revealed the absence of the Cu2p_3/2_ signal in the region close to 933 eV, which indicates that the purification process to remove the copper compounds of cycloaddition reaction was successful. The high-resolution (HRXPS) measurements showed the Au4f_7/2_ and Au4f_5/2_ doublet characteristic of the metal state and the absence of the AuI singlet at 84.9 eV (see Appendix A). The observed peaks shift to slightly higher energy values, as expected, due to the Au-S interaction. For sulfur, it was possible to observe the doublet corresponding to the S2p_3/2_ and S2p_1/2_ signals, however, as a consequence of the complexity of these signals.

### 3.4. Isothermal Treatment of the Gold Nanoparticles

In order to choose the best conditions for the isothermal treatments, the growth of the nanoparticles upon application of treatments at temperatures of 120, 150 and 180 °C in air atmosphere was investigated. These temperatures are below the decomposition onset (all the onset temperatures measured by TGA are higher than 270 °C, see above). This exploratory study was carried out on the nanoparticles of AuDT-AT-TA@L2-3,4. For each temperature, the sample was submitted to the thermal treatment for 30, 60, 120 and 180 min. The results confirm that at an intermediate temperature of 150 °C represents the most appropriate balance for an efficient growth without a significant loss of monodispersity in size or shape (see Figure 5 and Appendix A).

After these previous studies, samples of all the nanoparticles AuDT, AuDT-AT and AuDT-AT-TA@Ln were submitted to an isothermal treatment at 150 °C, a temperature at which the material is fluid but no decomposition or aggregation of the particles takes place, for 30, 60, 120 and 180 min in the presence of air and in the absence of solvents or stabilizing species. A summary of the data of the size of the thermally-treated nanoparticles drawn from scanning transmission electron microscopy (STEM) experiments are gathered in Appendix A. The progressive increase in the size of the nanoparticles as a function of the treatment time is assigned to a sequence of processes of nucleation, growing and Ostwald ripening [52,53,54].

As an example, in Figure 5 are gathered TEM microphotographs with their SAED patterns (a, c, e and g) and their respective nanoparticle size distribution histogram (b, d, f and h) of AuDT-AT-TA@L3-3,4,5 after 30, 60, 120 and 180 min of isothermal treatment at 150 °C. At t = 0 min the NPs have a mean size of 2.1 ± 0.3 nm. Upon thermal treatment, the particle size grows as the time increases, without aggregation processes and keeping, in general, a high degree of monodispersity in size and shape.

In addition, as has been observed in previous papers [32,35,37,40], a short-range arrangement of the nanoparticles in layers, or even in bilayers, is observed in some cases with nearly constant separation distances between the nanoparticles. Thus, in Figure 5i is represented a magnification of the image of AuDT-AT-TA@L3-3,4,5 (60 min), where it is possible to observe short-range interactions between nanoparticles with local hexagonal order. This observation is general for all the isothermal treatments of this compound. This fact probably reflects the tendency of the bare ligands to the liquid crystal order, although their derived nanoparticles are not mesomorphic.

Using the transmission electron microscope, the different nanoparticles AuDT-AT-TA@Ln prepared by cycloaddition and subsequent thermal treatment were studied by selected area electron diffraction (SAED). Similar to the results obtained in a previous work, sharp reflections were recorded that reveal the polycrystalline character of the samples. The interplanar spacings deduced from these reflections (see Appendix A) are consistent with a face-centered cubic lattice characteristic of metallic gold and the lattice constant *a* calculated from those reflections has the expected value, very close to the values published in the literature [55,56]. This parameter *a* remains constant regardless of the treatment time, and this suggests that the gold atoms do not undergo changes, such as oxidation, during this process.

The nanoparticles obtained after 180 min of thermal treatment were investigated by X-ray photoelectron spectroscopy (XPS) studies. Using this technique, it was observed that the samples stayed stable and decomposition did not take place, as revealed by the fact that the position of the signals characteristic of the S2p and Au4f practically remained unaltered respect to the initial values. Moreover, the signals tend to exhibit better resolution than for the non-treated nanoparticles (See Figure 5j,k, and Appendix A).

Taking into account the results obtained in the thermal treatment, and in order to facilitate the comparison between the pristine nanoparticles and the thermally-treated nanoparticles, in Table 3 are gathered the diameter, the ligand composition and the UV absorption data in solution of the pristine nanoparticles and the corresponding data of the nanoparticles treated at 150 °C during 60 min. This treatment time was chosen because of the homogeneity of the results obtained and the short-range arrangement of the nanoparticles in layers observed in these conditions.

#### 3.4.1. NMR Studies

The thermally-treated materials were studied by ^1^H-NMR to determine the percentage composition of each ligand (see Appendix A). The spectra do not show signals associated to aggregation of the metal cores or desorption of ligands in solution. It is important to take into account that the increase in the diameter of the AuNP’s is associated to a relative decrease in their surface, because the volume increases in a factor 3 (4/3 πr^3^), whereas the surface increases in a factor 2 (4 πr^2^). Logically, a significant increase in the volume of the NP’s can be associated to a loss of ligands because the decrease in the total surface of the whole sample prevents all of them from being accommodated. The results of the ^1^H-NMR studies confirm this fact. As noted above in the ligand exchange process, the AT ligand is the most likely to be removed, and in most cases the percentage of this ligand decreases after the thermal treatment. Particularly noteworthy is the case of AuDT-AT-TA@L6-3,4,5 nanoparticles in which the AT ligand practically disappears. This result has been confirmed in an additional study and could be a consequence of the larger particle size obtained in the thermal treatment of these nanoparticles (ɸ = 8.7 nm), which is associated with a very important decrease in the total surface. A consequence of this decrease in the percentage of the AT ligands is the corresponding increase observed in the percentage of DT ligand, whereas the percentage of the Ln (dendritic) ligands remain constant. Curiously, the behavior of the AuDT-AT-TA@L4-3,5 and AuDT-AT-TA@L9-3,4,5 nanoparticles is different and practically the ratio of the three ligands remain constant. It is important to take into account that these NP’s suffered a significant loss of ligands in the ligand exchange process, and therefore it is expected that the effect of surface reduction affects them much less markedly.

#### 3.4.2. UV-Vis Study of the Thermally-Treated Nanoparticles

The materials obtained by cycloaddition and subsequent isothermal treatment were studied by UV-vis spectroscopy in solution with the aim of detecting the surface-plasmon-resonance (SPR) bands of the nanoparticles. Figure 6 shows the variation in the wavelength maximum of the plasmon bands as a function of the treatment time for AuDT-AT-TA@L2-3,4 nanoparticles submitted to an isotherm at 150 °C (dashed line). It can be observed that, as a function of time, the surface plasmon resonance bands tend to become sharper and shift batochromically as the diameter of the metal core increases. The behavior of all the samples in this series of materials is the same (see Appendix A), and the wavelength maximum is affected by both the size of the metal core and the organic environment around the particle.

UV-vis spectroscopy has also been carried out in solid samples obtained by slow evaporation of the solvent from solutions in dichloromethane deposited on quartz. A noticeable shift of the absorption maxima was observed in all cases compared to those observed for the samples in solution (see continuous line in Figure 6). This shift increases as the mean diameter of the nanoparticles increases and can be as large as 52 nm. This result is mainly due to local interactions, as it is known that in the solid state the plasmon properties strongly depend on the interactions between the nanoparticles, which generate a near-field effect. This effect arises from the ability of adjacent nanoparticles to mutually couple their plasmon oscillation, which produces a batochromic effect and band broadening.

## 4. Conclusions

A series of polybenzylic dendrons with alkynyl focal point were synthesized. Five of the six alkyne focal point dendrons synthesized showed liquid crystal behavior of columnar type. XRD studies confirm that the dendritic units interact with themselves in order to generate disk-type assemblies and these disks stack in a columnar arrangement. In addition, the synthesis of gold nanoparticles with 11-azidoundecane-1-thiol has been successfully carried out by a ligand-exchange reaction and the subsequent coupling with the alkyne dendrons by [TBTA-Cu^I^]-catalyzed Huisgen azide-alkyne cycloaddition, giving place to hybrid nanomaterials. These processes occur without aggregation or decomposition of the metal core during the reactions and subsequent purification. The hybrid nanomaterials were carefully characterized and then thermally treated in mild conditions. Upon this treatment, the particle size grows as the time increases, keeping, in general, a high degree of monodispersity in size and shape. The best results regarding size homogeneity have been achieved for AuDT-AT-TA@L2-3. Compared with the gold nanoparticles described in our previous report [29], the nanoparticles described in the present work reach larger sizes, although with greater polydispersity. The thermal treatment leads to a modification of the percentage of each ligand attached to the gold surface. The most striking case is that of AuDT-AT-TA@L6-3,4,5 in which the AT ligand practically disappears. Dendron L4-3,5, with a V shape, shows a thermal behavior clearly different from that of the other dendrons, as it does not show liquid crystal properties. The nanoparticles bearing this dendron also show properties clearly different from those of the other nanoparticles; thus these NPs show a lower yield in the addition process and exhibit a phase transition between the solid state and the fluid phase at a temperature of 28 °C, significantly lower than those of the other nanoparticles studied.

A short-range arrangement of the nanoparticles in layers, or even in bilayers, is observed by TEM in some cases, with nearly constant separation distances between the nanoparticles. This probably reflects the tendency of the bare ligands to adopt a local hexagonal order, although their derived nanoparticles are not mesomorphic. Through UV-vis absorption measurements in solid state, this tendency to short-range order of the metallic cores has been confirmed. These results suggest the existence of near-field effect on the basis of the coupling of surface plasmon resonance band and bathochromic shift relative to the UV-vis results in solution. In conclusion, although the gold nanoparticles decorated with mesogenic dendrons described in this report do not exhibit liquid crystal properties, they meet the features for a remarkable self-assembly behavior.

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
