# Peer review of "Gold Nanoparticles Modification with Liquid Crystalline Polybenzylic Dendrons via 1,3-Dipolar Cycloaddition"

_nanomaterials, 2022, doi:10.3390/nano12224026_

Round 1
Reviewer 1 Report
The authors report on the preparation of gold nanoparticles (Au NPs) functionalised by liquid crystalline (LC) dendrons and their chemical-physical characterization. With respect to previous reports (e.g. the article ACS Omega 2021, 6, 1, 348–357), the dendrons are grafted on the particles thanks to a click reaction using azide groups already present on the particles. The use of a bifucntional ligand (containing thiol and azide) before the click reaction allows to limit the use of the dendron and can improve the functionalization degree.
The authors present many technical details that make the article not easy to read. Moreover, the general scope of the study should be better clarified: what are the importance of functionalising Au NPs with LC dendrons? To control aggregation (and thus optical properties), other simpler methods have been widely described.
I suggest to improve the article following the suggestions below:
1) The compounds should be named all times in the same way (e.g. not using Compounds 1, 2 and 5 in line 179 but always with abbreviation explained in Figure 1, or the opposite)
2) Paragraph 3.1 is very technical and difficult to follow. Can the authors summarize the effect on adding terminal alkyl chains or modifying the substitution position on the series? The authors should add some exemplificative POM images at least in the SI.
3) Lines 190-196 are repeated 2 times (eliminate lines 220-226)
4) Can the author study the LC properties of the final materials? They report some images in the solid phase where a short-range interaction between nanoparticles with local hexagonal order is highlighted. How the different dendrons modify the particle assembly? What is the effect of the temperatures on the different NP assembly?
5) The most critical part is to understand why the article reports on nanoparticles with many different dendrons without comparing their final properties. Why it is important to prepare and characterize theisset of nanomaterials?
6) What are the main differences (in term of final properties) with other NPs previously described (e.g. comparison with ACS Omega 2021)
Reviewer 2 Report
The paper is interesting and well-designed. Nonetheless, some aspects should be corrected including:
1) Title of the paper: "cycloadition" should be replaced by "cycloaddition". Next, the dot should be removed.
2) From editorial viewpoint, the text in the paper should be replaced.
3) Abstract of the paper should be supplemented with some quantified data. Moreover, the novelty of the research should be emphasized in this section.
4) The paper contains numerous abbreviations therefore it is strongly suggested to add additional subsection with all of them and their explanations.
5) The paper should be written using phrases such as "it has been maintained" instead of "we maintained..." (e.g. section 3.3.).
6) The discussion over the results obtained needs to be supplemented with references to other works.
7) Final conclusions should be supplemented with some quantified data.
8) Section References should be prepared in line with the requirements of the Journal, i.e. the whole journal names need to be replaced by their abbreviations.
Reviewer 3 Report
The manuscript submitted by Serrano et al. on gold nanoparticles decorated with dendrons is interesting work on integrating nanomaterials with liquid crystalline materials. The obtained nanostructures were fully characterized. I recommend publishing the work after minor changes.
The authors included mostly outdated references; please add some of the new references related to gold nanoparticles/dendrimers/liquid crystal. In addition, the two references at the bottom are recommended. They are related to gold-carbon/dendrimer nanostructures:
1. Chem. – Asian J., 2010, 5, 887–896
2. Nanoscale, 2017, 9, 3128-3132.
